# Adaptation after Extreme Flooding Events: Moving or Staying? The Case of the Ahr Valley in Germany

**Alessa Jasmin Truedinger \*, Ali Jamshed, Holger Sauter and Joern Birkmann**

Institute of Spatial and Regional Planning, University of Stuttgart, Pfaffenwaldring 7, 70569 Stuttgart, Germany

\* Correspondence: alessa-jasmin.truedinger@ireus.uni-stuttgart.de; Tel.: +49-711-68566329

**Abstract:** More than 130 lives were lost in the 2021 heavy precipitation and flood event in the Ahr Valley, Germany, where large parts of the valley were destroyed. Afterwards, public funding of about 15 billion Euros has been made available for reconstruction. However, with people and settlements being in highly exposed zones, the core question that is not sufficiently addressed is whether affected people want to rebuild in the same place, or rather opt to move out. The paper explores this question and assesses motivations and reasons for moving or staying in the Ahr Valley. For this purpose, a household survey was conducted focusing on 516 flood-affected households. The collected data was analyzed using descriptive and inferential statistics. The results revealed that the ownership of the house or flat significantly influenced the decision of whether to stay or to leave. In addition, an attachment to the place and the belief that such extreme events occur very rarely influenced the decision to stay and rebuild. Age, gender and household income barely influenced the decision to stay or to move to a new place. Interestingly, results demonstrated that many respondents view settlement retreat and the relocation of critical infrastructures as important options to reduce risk, however, many still rebuild in the same place. These insights enable local policy and practice to better address the needs of the population in terms of whether to stay or move after such an extreme disaster.

**Keywords:** relocation; adaptation; extreme events; reconstruction; Ahr Valley; floods in Germany 2021

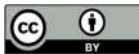

## 1. Introduction

Relocation and migration after extreme events are discussed both in the context of disaster risk reduction and in the context of climate change adaptation [1,2]. In this regard, the strategy of relocation is taken up especially in international frameworks. Thus, planned resettlement is seen as a possible cross-sectoral adaptation option in the contribution of the second working group to the Sixth Assessment Report of the Intergovernmental Panel on Climate Change [3]. The Sendai Framework, in turn, recommends, on the one hand, creating the political framework for relocating settlements from risk-prone zones—applicable throughout the entire disaster management cycle—and, on the other hand, relocating public facilities and infrastructures from the risk-prone area precisely in the reconstruction process in the sense of "Build Back Better" [4]. The German Strategy for Adaptation to Climate Change, which is applied at the national level, and its detailed monitoring report also mention avoiding settlement development in areas with climate hazards, but do not yet speak of settlement withdrawal and relocation [5,6].

Relocation and migration can reduce exposure, potentially also reduce vulnerability and increase resilience [7]. However, this also depends on whether and how the relocation is planned and carried out and how new locations are characterized [7,8]. Significant displacement has occurred to date in the context of floods. Between 2008 and 2020, 49% of all disaster-related displacement was due to flooding—encompassing about 156 million

people [9]. This problem is getting more severe in the future, since with each degree of temperature increase, the risk of flood-related displacement increases significantly by more than 50% [10]. An increase in heavy rainfall events and flooding is expected in many regions around the world [11]. In particular, it is expected that flood risk increases in almost all Western and Central European countries [12]. However, the projections regarding the increase or decrease of heavy rainfall events in Germany are subject to great uncertainty and different development directions are possible for different regions. For example, over the last seventy years, the frequency of heavy rainfall events has increased in parts of southern and northern Germany, while it has tended to decrease in central Germany [13]. Nevertheless, the intensity of heavy rainfall events in Germany has increased so far [13]. Attribution studies have also shown that the heavy rain events that led to the 2021 flood have become more intense and more likely due to climate change [14]. Thus, preventive risk reduction and adaptation are more essential than ever.

Relocation is likely to gain further attention as a transformative measure to cope with and adapt to climate-influenced extreme events [8,15]. Relocation after extreme events is often forced by state authorities, which is why most of the scientific literature focuses on such involuntary relocation processes [16–18]. Planned, strategic relocation has also been studied, although less intense [19]. However, only limited literature and very few systematic studies exist on how people affected by extreme events like floods view and decide on relocation, migration, and settlement retreat, and on whether people aim to move temporarily or permanently out of the exposure zone [20,21].

In this regard, the paper provides new and innovative insights into how people view relocation and migration after the major Ahr flood disaster of July 2021, and on factors that are decisive for affected people in choosing a new location. The devastating flood event in Western and Central Europe in July 2021 destroyed a large number of buildings and made many people (temporarily) homeless, particularly in the Ahr Valley in Germany [22]. Against this background, we conducted a household survey to explore whether and why people affected want to stay or move. The household survey, undertaken between June and August 2022 in the Ahr Valley with 516 respondents, provides new data and important insights into this complex topic. These findings can also help to guide and modify reconstruction policies, including issues of relocation and the development of alternative settlement sites.

## 2. Flood Impacts in the Ahr Valley and Perspective on Reconstruction and Relocation

The heavy rains that fell in Western and Central Europe in mid-July 2021 resulted in severe and sudden flooding especially in Belgium, the Netherlands, and Germany, which was hit particularly hard. In Germany alone, more than 180 deaths [23] and damages amounting to 33 billion euros [24] were recorded. The losses were almost exclusively concentrated in the two German states of North Rhine-Westphalia and Rhineland-Palatinate, with the Ahr Valley in the latter achieving particular sad notoriety. Over 70% of all fatalities in Germany occurred in the Ahr Valley, where entire houses were washed away and where villages were completely destroyed [23,25].

There are several reasons why such aforementioned extreme destruction occurred. On the one hand, the Ahr Valley is a typical low mountain region with steep slopes and narrow valleys, which has been cultivated and inhabited extensively by people for a long time [26,27]. Therefore, due to the confined space, a large number of people and buildings are located in exposed areas. In addition, such regions are typically prone to mass movement, fast and erosive discharge, and high debris [27]. The latter led to severe clogging and subsequent destruction of many of the 75 bridges in July 2021, further increasing the flood surge of the Ahr river [27]. On the other hand—and this is now again the case for the entire affected region—the soils were already saturated by previous rain events. Thus, the almost stationary, heavy rainfall from 12–15 July, whose meteorological driver was the low-pressure system "Bernd", contributed virtually exclusively to the runoff event, and even smaller inflows became raging rivers [27].

Next to the high exposure of people and settlements along the Ahr, the high vulnerability of the population has also played a major role in terms of the severity of losses and damages observed. Almost 80% of the fatalities were older than 60 years [25]. The average age of the population in the county of Ahrweiler is in the upper quarter of the counties in Rhineland-Palatinate with about 46.7 years [28]. The city of Bad Neuenahr-Ahrweiler, for example, has significantly more people over the age of 65 (31.2%) than other municipalities of the same size where on average only 23.7% of the population is over 65 (as of 31 December 2021) [29]. In addition, a large number of critical and sensitive infrastructures are also located in the floodplain. In Sinzig, a city downstream the Ahr, twelve people from a residential care home for people with disabilities died due to the fast increase of the water level and the lack of effective early warning and preparedness measures [30].

In light of high exposure and high vulnerability, the option of relocation of people and settlements is a hot topic within the reconstruction process. However, Greiving et al. already underscored that, especially in industrialized countries, relocation and settlement retreat are often seen almost exclusively as the last option—and can be legitimized in particular when comprehensive flood protection measures require a disproportionate amount of money [31]. Though it is rarely conducted, it is still considered in Germany as a measure to cope with extreme events and it has also been conducted for development projects (e.g., coal mining) [31]. In terms of climate change adaptation and disaster risk reduction, there are only a few, mostly isolated examples in Germany [32]—since German regional planning and building law is almost entirely designed to control and implement settlement and infrastructure growth instead of dismantling such structures [33]. In Germany's neighboring country, Austria, which is quite similar to Germany in terms of population, politics, and administration, there was one example of such a settlement retreat in 2016 in the Eferding Basin ("Eferdinger Becken"), initiated by several floods and based on a voluntary manner [34,35].

Even though settlement retreat and relocation in the context of climate adaptation and risk reduction after extreme events are solely implemented in very few cases in Germany [31], there is still a great need for further research in this area, as not only the planned resettlement but also the individual decisions of people to stay or move after extreme events need to be better understood. Hence, there is a need for, on the one hand, an improved understanding of the acceptance of relocation strategies by those affected—including their needs and the support provided by public institutions—and, on the other hand, a better understanding of individual decisions regarding staying or out-migrating. Post-disaster processes, therefore, offer an important opportunity to examine such questions.

It is precisely during the reconstruction phase that questions about relocation and migration occur. In addition, reconstruction processes also require the assessment of whether existing houses and settlement structures should be rebuilt in the same place or whether they need to be dismantled. For example, in Germany, the protection of the status quo can be undermined if the event has led to a complete destruction of the former house or company. Moreover, such situations also allow revisiting past policies and might create an atmosphere where "new approaches" are developed and tested. Against this background, it is particularly interesting to investigate the attitude of people at risk towards moving or staying at the site as well as settlement retreat after such a disaster. As part of the KAHR project, an extensive household survey was conducted in the aftermath of the July 2021 flood disaster in the county of Ahrweiler in order to explore these issues and to assess different types of impacts, mental stress, (prevention) measures, and reconstruction processes. Particular attention was given to issues of settlement retreat and relocation. In this context, the following research questions are addressed in this paper:

- How do affected people in the Ahr Valley assess relocation as well as settlement retreat?

- What types of relocation and migration can be observed (e.g., temporarily or permanently)?
- Who relocated permanently from the area and who did so only temporarily or not at all?
- What motivates affected people to stay or to leave the original location?
- Which factors are decisive for people moving out when choosing a new location?
- How do people evaluate the need for relocating critical and sensitive infrastructures?

## 3. Materials and Methods

### 3.1. Case Study Area

The county of Ahrweiler is located in the German state of Rhineland-Palatinate, on the border with North Rhine-Westphalia (see Figure 1). Its area is 787.03 km² and it is home to about 130,000 inhabitants, of which 49.4% are male and 50.6% are female, and one tenth of the population is foreign [36]. The county is mostly rural, with small municipalities united into so-called associated municipalities. However, there are also several independent cities, of which Bad Neuenahr-Ahrweiler is the largest with about 26,500 inhabitants [29]. The gross domestic product of the county in 2019 was about 3.56 billion euros [36]. More than 9000 mainly small and medium-sized enterprises from the economic sectors of trade, tourism, crafts, industry and services are located in the county of Ahrweiler [37]. The county is characterized by viticulture and tourism and therefore serves as a recreation area for the major city of Bonn.

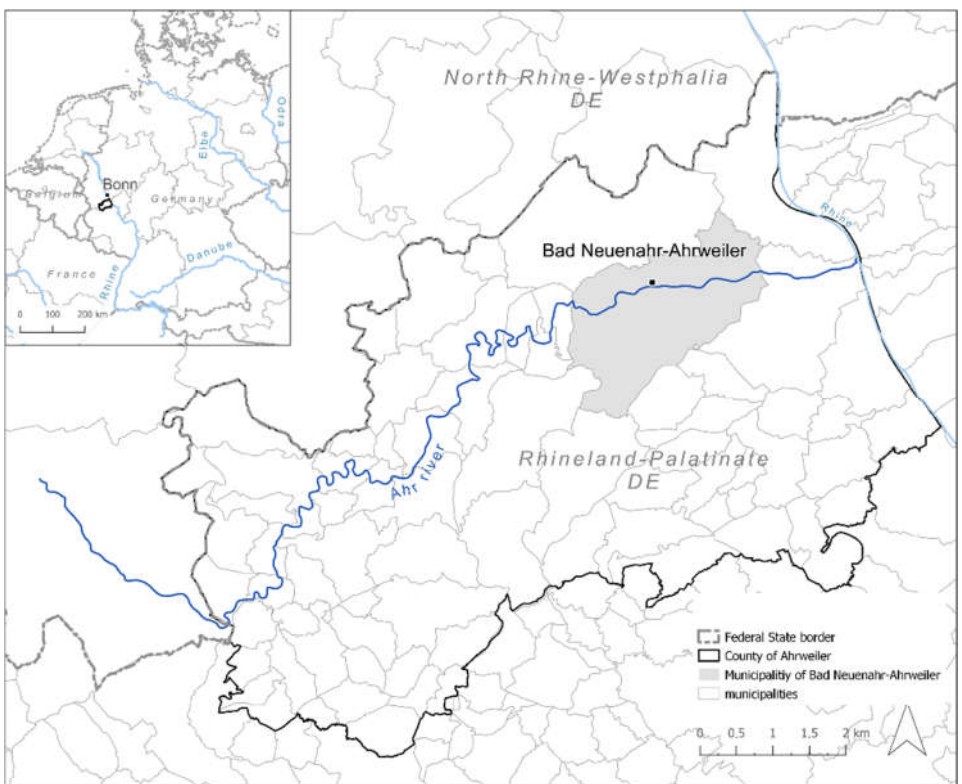

**Figure 1.** Overview map of the study area. Authors own illustration based on data of the German Agency for Cartography and Geodesy—Bundesamt für Kartographie und Geodäsie (BKG Bund): administrative boundaries and rivers: © GeoBasis-DE/BKG, 2022.

The Ahr Valley, which includes the county of Ahrweiler in particular, is part of the Paleozoic Rhenish Massif [38]. The geological formations were formed about 400 million years ago by the deposition of clay shales, siltstones, banded shales and sandstones in

constant alternation [38]. The region is named after the river Ahr, which characterizes the entire landscape, whereby the Ahr also has several important tributaries. The source of the Ahr is in Blankenheim, North Rhine-Westphalia; however, 68 km of the total 86 km length of the river is in Rhineland-Palatinate [39]. 76% of the precipitation catchment area of the Ahr and thus 680 km² are also located in Rhineland-Palatinate [39]. The average annual precipitation level is rather low at 675 mm—only the flood origin area in the High Eifel is high in precipitation [39]. Nevertheless, the high precipitation, the spring tide-like swelling of the Ahr and the subsequent catastrophic floods were not new in 2021. Already in 1804 and 1910, the Ahr Valley experienced exactly the same processes—and both disasters caused more than 50 casualties each [26]. Of the approximately 56,000 people living along the Ahr River in 2021, 42,000 were affected by the July 2021 flood disaster—and 133 lost their lives [22,25]. At least 17,000 people were left with almost nothing after the disaster, and more than 9000 buildings were completely destroyed or severely damaged [22]. In addition to private homes, many schools, kindergartens, nursing homes and hospitals were also affected, as were sewage treatment plants and power infrastructures.

### 3.2. Collection and Description of the Household Survey Sample

Since the dimensions of this disaster were unprecedented in Germany for a long time and no primary data were available, it was essential to conduct a household survey to better understand the views of those affected. In order to participate in the household survey, 5250 people in the county of Ahrweiler, who had applied for emergency aid ("Soforthilfe") after the flood disaster, were contacted in June 2022 with the help of the county authorities. About 30–40 letters could not be delivered by the post office, because the persons could no longer be found at the reported location and no contact tracing was available. A total of 516 people, and thus 9.9% of those contacted, took part in the survey between June and August 2022. The survey was mostly conducted online using the EvaSys survey software, with only 21 people completing a paper questionnaire. The option of a printed questionnaire was provided especially to reach the many elderly people as well. The only requirement for participation was a minimum age of 18. Some basic information about the sample is presented in Table 1.

**Table 1.** Basic information about the sample and the population—for the latter, the information originates from official statistics [36].

|  | **Sample** | **Official Statistics** |
| --- | --- | --- |
| Age (0–19/20–64/65+ years) [%] | -/67.0/33.0 | 17.6/57.4/25.0 (-/69.7/30.3) * |
| Gender (male/female/diverse) [%] | 52.4/47.6/- | 49.4/50.6/- |
| Income | 2600–3599€ per household (median) | 2030€ per resident |
| Homeowners [%] | 67.6 ** | 52.5 *** |

* by removing the 0–19 year olds from the official statistics ** including owned by close relatives, valid in July 2021 *** data valid for Rhineland-Palatinate in 2006 [40].

Since only those who had applied for emergency aid ("Soforthilfe") were contacted and no one under 18 was allowed to participate anyway, the group of minors is not present. All participants were even over 20 years old. If one also removes the under-20s in the official data, as it is done in Table 1, it can be concluded that the sample is representative in terms of age. It is also representative in terms of gender, although males are slightly more represented in the sample than females and than official data indicate. A potential reason for this could be that in households with conservative role models, males may have tended to be more likely to apply for emergency aid. At first glance, income appears to be significantly higher among survey respondents, but this is due to the fact that the income of the entire household was queried, and the official data indicate household income per inhabitant. The proportion of owners is also very high in the

sample—but the cases in which people lived on the property of close relatives in July 2021 are also counted in the case of the sample.

*3.3. Analysis Framework*

The survey method is a quantitative household survey with a standardized questionnaire (Supplementary Materials) that partly builds on previous surveys of the Institute of Spatial and Regional Planning and the Institute of Environmental Sciences and Geography in terms of content. In this context, on the part of the Institute of Spatial and Regional Planning, the study of Weißer et al. [41] should be emphasized. Most of the questions were formulated in closed form, although some questions with free-text answers were also embedded. Both multiple choice and single choice were included, as well as dichotomous and Likert scale questions. In order to protect privacy and avoid arbitrary ticking, there was an option to select "no answer" for each question—but this had to be indicated compulsorily in order to further process the questionnaire. The questionnaire was pretested prior to the real household survey by having it edited by several project staff from on-site who were themselves affected.

To answer the research questions, various questions from the survey were analyzed using statistical methods through the IBM Software SPSS (originally abbreviation for "Statistical Package for the Social Sciences"), version 28.0.0.0 (190). Frequency analyses, cross-tabulations and Spearman's correlation were used [42]. The effect size is classified as weak, moderate, and strong according to Cohen [43]. In the case of Spearman's correlation, the coefficient corresponds to the effect size; thus, the limits are at $\varrho = 0.1$ (weak), $\varrho = 0.3$ (moderate), $\varrho = 0.5$ (strong) [44]. To examine correlations between reasons for staying, for moving, for choosing a new location, and various variables such as age, gender, or household income, Spearman's correlation was used. For a better understanding, we have refrained from using the statistically correct procedure in every case. For instance, in the case of two dichotomous variables, the phi coefficient would actually be correct from a statistical point of view and not Spearman's correlation, but in that case, one arrives at exactly the same result. Also in the case of a relationship between a dichotomous and an ordinal variable, Spearman yields a good result, which is better to interpret than e.g., the chi-square test. For better clarity and readability, not all correlation coefficients and significances are directly mentioned in the text—however, these can all be found in Appendix A in Table A2. All variables used and their frequencies or means plus standard deviations are also listed in Appendix A in Table A1.

The survey was approved by the "Kommission Verantwortung in der Forschung" (Ethics Committee) of the University of Stuttgart (Ref. 22-017, 6 July 2022).

**4. Results**

The survey results provide new insights into how people affected view relocation versus staying in the same place. We explored whether there are certain groups of people who are more likely to decide for or against relocation. In addition, motivations that influenced these decisions were captured and examined along different age and income groups as well as genders.

*4.1. Housing Situation and Relocation Behavior of Different Groups*

Overall, 41.9% of respondents had to leave their house/apartment after the 2021 flood at least temporarily. 31.0% of them were able to return within two months, 32.9% between two and eight months, and 36.1% of them were not able to return by the time of the survey. In total, 14.1% of respondents had already moved permanently one year after the event—and more than half of those had moved to another municipality (see Figure 2). Most respondents, 73.0%, are living in the same house or apartment one year after the flood disaster. However, it is important to note that this does not mean that their house or apartment has already been fully renovated or refurbished. In some cases, for example,

people are only living on the second floor of their house. About 13% of the people interviewed are still living in temporary accommodation or with friends or relatives, one year after the disaster. That means they are not yet able to return to their old place of living. Out of the people who still live with friends or in temporary houses, about 14% plan to move permanently into another location. This percentage is quite different from the group that is still living in the same building. In that group, only 6% plan to move or migrate out of the former living location (see Figure 2).

**Housing situation one year after the event**

**Figure 2.** Housing situation one year after the event, $n = 512$.

This correlation between the current housing situation, i.e., in the same building or in temporary housing, and the decision for or against a future move is also statistically significant. People living in the same building plan to move less often than people living in temporary accommodation, Spearman's ϱ = −0.116, $p = 0.018$, although the correlation is rather weak.

As can be seen in Table 2, over half of the respondents who had already moved permanently and could provide information on the location of the old as well as new residence had moved completely out of the July 2021 flood zone. For example, 50% of the people who lived within the originally designated floodplain in July 2021, now (as of August 2022) live completely outside the July 2021 flood zone, and 66.7% of the people who already lived outside the legally designated floodplain but within the July 2021 flood zone. However, the sample size is rather small.

**Table 2.** Cross-tabulation showing from which area to which area the respondents have already moved permanently, $n = 45$.

| After the 2021 Flood→ Before the 2021 Flood ↓ | Inside the Legally Designated Floodplain Valid in July 2021 | Outside the Legally Designated Floodplain Valid in July 2021, but within the July 2021 Flood Zone | Outside the Legally Designated Floodplain and Outside the July 2021 Flood Zone |
|---|---|---|---|
| Inside the legally designated floodplain valid in July 2021 | 4 | 3 | 7 |
| Outside the legally designated floodplain valid in July 2021, but within the July 2021 flood zone | 2 | 8 | 20 |

| Outside the legally designated floodplain and outside the July 2021 flood zone | 0 | 0 | 1 |
|---|---|---|---|

Since some people were still planning to move at the time of the survey and it made no difference in terms of statistical significance whether they were added to those who had already moved, this was done when examining different groups in terms of moving behavior. With regard to the question of staying or moving, we formed two groups of tenants and owners (including owned by close relatives). The relationship between ownership and staying and respectively tenancy and moving is particularly striking. There is a moderate but significant correlation between ownership and future intended housing situation one year after the event, $\varrho = -0.323$, $p < 0.001$, i.e., renters are statistically significantly more inclined to move than owners. This can also be seen Figure 3, which shows the distribution of staying, planning to move, and moving carried out, calculated down to the tenants and owners who participated respectively. While 60.5 % of renters surveyed want to stay on site and 39.5% of those are planning or have already carried out a move, 88.4% of owners (including those whose building/apartment is owned by close relatives) surveyed want to stay and thus only 11.6% have planned or carried out a move.

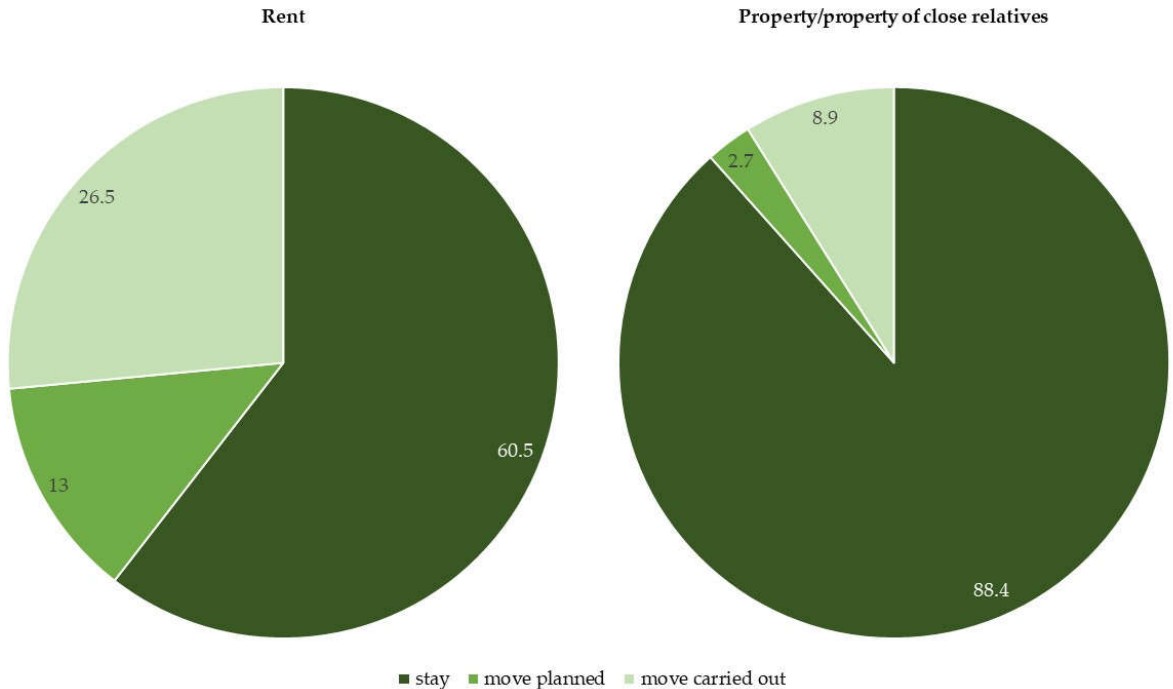

**Figure 3.** Distribution of tenants as well as owners surveyed on staying, moving planned and moving carried out, *n* = 162 (rent) and *n* = 327 (property/property of close relatives).

In addition to ownership, it is also interesting to investigate several other variables and groups, mostly sociodemographic, such as gender and age, as independent variables in regard to the future intended housing situation. However, neither age nor gender nor household income has a statistically provable influence on the future intended housing situation. Only the currently still existing damages and the location of the original residential building could potentially have an influence, as a statistically significant relationship is discernible in this respect.

Gender is not statistically significantly related to the future intended housing situation, $\varrho = 0.034$, $p = 0.452$. Furthermore, age, as a sociodemographic factor, could

influence the decision to stay in or leave the July 2021 residence. Again, there is no statistically significant correlation between the five age groups and relocation behavior, $\varrho$ = −0.071, *p* = 0.117. Another factor that was examined is the influence of net household income. Interestingly, no statistically significant correlation between household income and the future intended housing situation was found, $\varrho$ = −0.067 and *p* = 0.167. Even when further distinguishing the current housing situation for the group that has not yet permanently moved, there is no statistically significant correlation between net household income and living in the same house/apartment as in July 2021 vs. living in temporary housing.

Moreover, we examined whether a statistical relationship between staying or moving and the damage experienced could be found. In this regard, nine different damage classes were applied—ranging from under €500 to damages of €100,000 or higher. The median is 100,000€ or higher, as 255 out of 492 respondents indicated this damage class. No statistically significant correlation is evident in this case either. On the other hand, there is a statistically significant correlation between the current condition of the building/property compared to before the flood on a six-point Likert scale and the net household income, $\varrho$ = −0.154 and *p* = 0.002. That means, the higher the household income, the more likely it is that the damage has already been replaced. And there is another statistically significant relationship between the current condition of the building/property compared to before the flood and the future intended housing situation, $\varrho$ = 0.118, *p* = 0.011, whereby people are more likely to relocate if the damages are still substantially (note: but these correlations both have a weak effect size).

In addition, if looking at respondents who have not yet moved permanently, the analysis showed that there is a statistically significant weak relationship between the current damages and living in the same house/apartment resp. living in temporary housing, with $\varrho$ = 0.293, *p* < 0.001. In this case, individuals tend to live more often in temporary housing and less often in the same building when damages are still more severe. However, ownership has no influence on current residence in this regard (i.e., same building or temporary housing).

Furthermore, it can be assumed from previous studies that it also plays a role whether one is a newcomer or a long-time resident [35]. Therefore, the household survey also questioned people about how long they had lived in their house/apartment of July 2021. In total, five categories were formed for the purposes of this study. In this respect, respondents move slightly more often if they have lived in the house/apartment for a relatively short time (0 to 5.5 years) and they stay in the house/apartment more often if they have lived there for a long time (10.5 years or more). This relationship is even statistically significant, albeit with a weak effect size, as $\varrho$ = −0.139, *p* = 0.002.

Moreover, correlations between staying/moving (both planned and carried out) and risk awareness and, concomitantly, the location of the original residences were also examined. In this context, it became apparent that the vast majority of respondents did not know that they lived in a flood-prone area before the 2021 event (see Figure A1a). This pre-flood risk awareness is statistically related to the location of the residential building and the location of the residential building is again statistically significantly related to the desire to stay resp. to move. Nevertheless, caution should be taken when interpreting these relationships. A total of 401 people responded to the question about the location of their original place of residence in July 2021. For the purpose of simplicity and interest, two main categories were formed: within the legally defined floodplain valid in July 2021 (note: in Germany, in particular, areas where, according to statistics, a flood occurs once every 100 years) and outside of it. The fact that 115 people did not provide any information suggests that some people are actually unaware of whether their residential building of July 2021 was located inside or outside of the legally defined floodplain. 25.7% stated that the building they resided in July 2021 was located within the legally designated floodplain at that time. Thus, of 74.3% of the respondents, the building was located outside the legally designated floodplain in force in July 2021. 496 respondents also provided information on

their pre-flood risk awareness. Of these, 17.7% knew they lived in a flood-prone area before the July 2021 event (see Figure A1b). Interestingly, there is a correlation between the place of residence (inside or outside the originally legally designated floodplain) and risk awareness prior to the 2021 flood (see Figure A1b). This shows, with $\varrho = 0.214$ and $p < 0.001$, that people who lived within the legally designated floodplain in July 2021 were more often aware of flood risk before the flood than people outside this area. However, pre-flood risk awareness is not statistically significantly related to staying or moving (both carried out and planned). In contrast, place of residence (inside/outside the legally defined floodplain valid in July 2021) is statistically significant, but weakly, related to staying or moving (carried out and planned), $\varrho = -0.143$, $p = 0.005$. Thus, people are more likely to stay in their original residence if they live outside the 2021 legal floodplain, which could be due to the fact that houses within the legal floodplain are usually closer to the rivers. On the other hand, it could also be related to the fact that from the sample, significantly more renters lived inside the "old" legally defined floodplain of July 2021, whereas more owners lived outside this area, $\varrho = 0.313$, $p < 0.001$. That it is not the other way around, and that tenure does not matter but only location, can be justified by the fact that living for rent has been mentioned by a large number of those who had moved or wanted to move as a decisive reason (see below).

### 4.2. Reasons for Relocation and the Selection of a New Location

In summary, 102 respondents indicated that they had either already moved permanently (72 persons) or were planning to do so (30 persons), which corresponds to 19.8% of respondents. 22 other participants made no indication, which is why a total of 124 people were able to answer the questions on the reasons for moving and the factors influencing the decision for a new location. In the vast majority of cases, the relocation is voluntary, as only two individuals indicated that their buildings were in the special hazard area ("yellow zone") and could not be refurbished, meaning that reconstruction for these buildings is prohibited by the state and relocation must necessarily take place. The reasons for relocation are shown in Figure 4, where multiple choices could be mentioned. Almost half of the 124 respondents said that living in rented accommodation was a reason for their move or desire to move, which is consistent with the fact that proportionately more renters have moved or want to move compared to people that own a flat or house. Almost the same number of people stated that their place of residence no longer offered the quality of life they expect. Slightly more than a quarter also mentioned that the location is too dangerous and therefore they opt to migrate out of this place. In addition, about 10.5% of the respondents mentioned as a reason for migrating out that the reconstruction was too expensive or too complex. This point was stated least frequently, but nevertheless, one tenth of the respondents are unable to cope with reconstruction financially, in terms of time or in terms of expertise and skills. It is also interesting to examine which reasons play a particular role for which population groups in terms of moving or staying. Age, gender and household income were again considered for this purpose. This differentiation might also inform future reconstruction policies.

**What are the reasons for your relocation?**

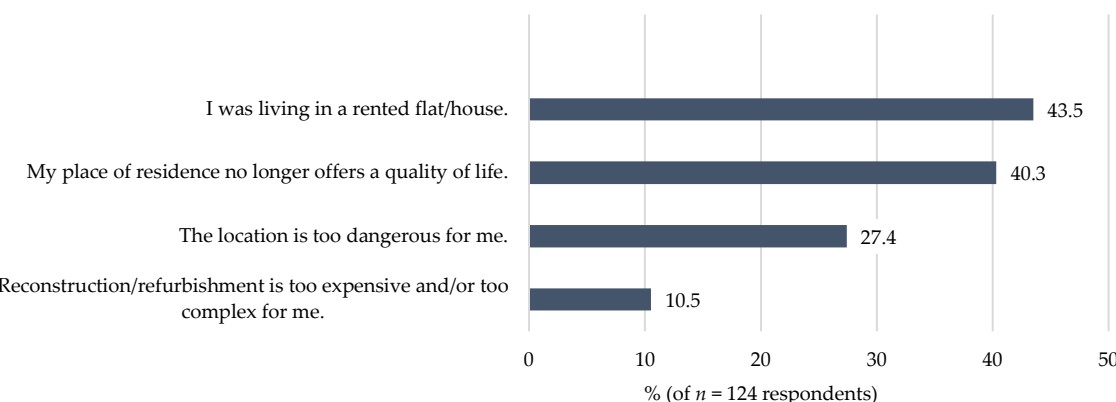

**Figure 4.** Reasons for relocation, $n$ = 124.

The analysis of age distribution and reasons for relocation (see Figure 5) provides additional insights. For example, it can be noticed that in the young age group of 20 to 29 years, a tenancy was given as a reason frequently—and most frequently in relative terms. In the 30 to 49 age group, tenancy also plays quite a large role. The fact that the place of residence no longer offers a quality of life appears to be equally decisive for all age groups, although this reason seems to play a slightly greater role among young adults as well as people of retirement age. If one considers the reason for the original location being too dangerous, the very elderly aged 80 and over in particular disproportionately often gave the danger at the original location as a reason in relative terms, but there are hardly any differences between the other age groups. In the 20 to 29 age group, too high complexity and/or too high cost of reconstruction or refurbishment were still selected most often in relative terms, although this reason was rarely ticked overall.

**What are the reasons for your relocation?**

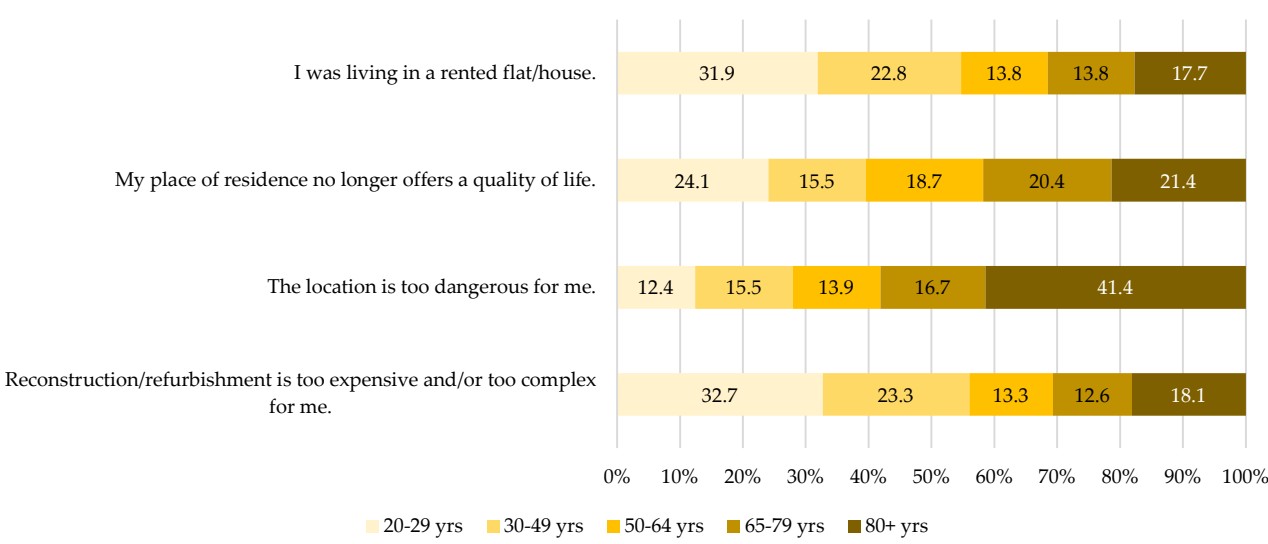

**Figure 5.** Reasons for relocation by age group, calculated down to participants from the respective age groups with $n$ = 122.

However, only in the case of "I was living in a rented flat/house" does a significant correlation according to Spearman emerge at a significance level of 5%, with $\varrho = -0.215$, *p* = 0.018. Yet, for the location being too dangerous, the expected and observed frequencies for the highest age group nevertheless show that substantially more 80-year-olds and older gave this reason than expected. Since there cannot be a monotonic relationship in the case of the reason of "no quality of life", because the two edge groups have the highest values, the Kruskal-Wallis test is actually more suitable for this reason. However, this also provides that there are no significant differences between the different age groups with regard to this reason, H = 1.274, p = 0.866. Moreover, the counts for the reason "reconstruction/refurbishment is too expensive/complex" are too small to obtain statistically robust results.

In addition to age, the influence of gender on the selection of reasons for moving was also investigated. But there is no significant correlation at all between gender and the choice of reasons for moving. We also examined correlation between net household income and the choice of reasons, since different reasons may be relevant in the case of different financial conditions. But here, too, there is no statistically significant correlation.

Apart from the reasons for relocation, it is also relevant to understand which factors are decisive for the choice of a new location. Figure 6 shows the importance of various factors in respondents' decision to locate a new site on a six-point Likert scale. About three quarters of all respondents (76.4%) consider a good social environment to be important or very important, so the mean value (mv) of 5.11 for this factor is the highest. Almost equally important is a flood-proof location for the new building (mv = 5.09). Good access to local supply is also rated as important or very important by 73.4% of respondents. And well over half consider sustainable and energy-efficient construction of the new building (67.3%) as well as financial support from the state (60%) to be important or very important. The importance of good public transport connections, a flood-adapted design of the new building, a short distance to the original place of residence and the possibility of remaining in the same (associated) municipality follow in descending order of importance regarding the mean value. Although the short distance to the former place of residence and remaining in the same (associated) municipality have a rather low mean value compared to the other reasons, they are still (very) important to 42.4% and 44.9% of respondents, respectively. The factor of a joint move with other affected neighbors is far behind, with only 11.3% considering it important or very important and 73.2% considering it unimportant or completely unimportant.

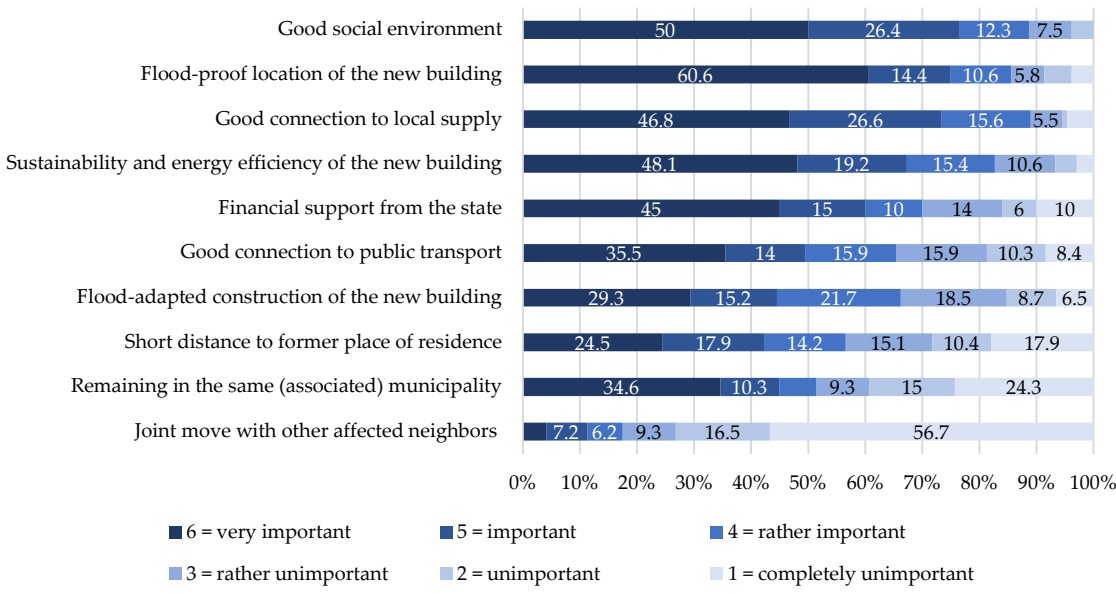

**Figure 6.** The importance of various factors in respondents' decision to locate a new site on a six-point Likert scale, sorted by the mean values (note: for clarity, all percentages less than 5% are not labelled).

Again, it is examined whether there are differences between the different age groups in the importance of potentially determining factors for choosing a new location. This is examined using Spearman's ϱ—for each factor except one, the significance clearly exceeds the 5% level, as can be seen in Table 3, and the null hypothesis that there is no correlation between the different age groups in indicating the importance of the factors listed cannot be rejected. Solely in the case of the factor "sustainability and energy efficiency of the new building" a statistically significant correlation between age and the indication of the importance of this factor can be found. The older the respondents are, the more important the factor "sustainability and energy efficiency of the new building" is to them when deciding on a new location.

**Table 3.** Spearman's ϱ for different age groups/ genders/ net household income classes and the importance of several factors when deciding on a new location.

| | Age | | | Gender | | | Household Income | | |
|---|---|---|---|---|---|---|---|---|---|
| | **n** | **ϱ** | *p* | **n** | **ϱ** | *p* | **n** | **ϱ** | *p* |
| Good social environment | 105 | −0.052 | 0.595 | 104 | −0.187 | 0.058 | 90 | 0.064 | 0.550 |
| Flood-proof location | 103 | 0.109 | 0.274 | 102 | −0.129 | 0.195 | 88 | 0.026 | 0.813 |
| Good connection to local supply | 108 | 0.033 | 0.738 | 107 | −0.087 | 0.371 | 93 | 0.180 | 0.085 |
| Sustainability and energy efficiency of the new building | 103 | 0.232 | 0.019 | 102 | −0.084 | 0.401 | 88 | −0.052 | 0.627 |

| | | | | | | | | | |
|---|---|---|---|---|---|---|---|---|---|
| Financial support from the state | 99 | −0.065 | 0.521 | 99 | −0.008 | 0.936 | 87 | −0.247 | 0.021 |
| Good connection to public transport | 106 | 0.163 | 0.095 | 105 | −0.014 | 0.887 | 91 | 0.163 | 0.095 |
| Flood-adapted construction of the new building | 91 | 0.121 | 0.251 | 90 | −0.235 | 0.026 | 91 | 0.189 | 0.073 |
| Short distance to the former place of residence | 105 | −0.104 | 0.292 | 104 | −0.200 | 0.042 | 91 | 0.050 | 0.641 |
| Remaining in the same (associated) municipality | 106 | 0.022 | 0.819 | 105 | −0.216 | 0.027 | 92 | −0.016 | 0.881 |
| Joint move with neighbours | 96 | 0.012 | 0.906 | 95 | −0.027 | 0.793 | 81 | −0.101 | 0.368 |

Furthermore, we also examined whether there are differences concerning gender (male and female) in indicating the importance of factors when choosing a new location. Here, Spearman's correlation shows that females significantly place greater importance than males on the short distance to the former residence, on remaining in the same (associated) municipality, and on a flood-adapted design of the new building (see Table 3).

Even though women in this case place a greater importance on the flood safety of the building, which can provide an additional level of safety when already moving to a presumably safer area, the women surveyed did not place a greater value on a flood-proof location (see Table 3) as well as on safety in general. The respondents were asked to indicate on a six-point Likert scale how much they agreed with the statement that they are someone to whom it is important to live in a safe environment and who avoids anything that endangers their own safety. According to Spearman's correlation, there is no difference in agreement with the statement between men and women. However, in terms of flood risk and the severity of the impacts of a potential future event, men and women again differ. Women assumed it is more likely that their current house/apartment will be affected by a flood again. Women also estimated the negative consequences of a possible event to be worse. All in all, for the women affected and interviewed, the flood risk, the negative consequences of a future flood event, and thus the desire for flood safety, even in a new location, seems to be more present.

Finally, we examined whether there is a relationship between net household income as an economic factor and the importance of factors that could be decisive in the choice of a new location by using Spearman's ϱ (see Table 3). Here, a significant correlation is only visible in the case of the factor "financial support from the state". The higher the net household income, the less important this factor is in the decision for a new location.

*4.3. Reasons for Staying*

In the same way that those who had already moved or planned to move were examined in more detail, so were those who had decided to stay. Two reasons clearly emerged when considering the reasons for staying in the same place of residence—both being internal factors (see Figure 7). At 56%, the most frequently cited reason is that respondents feel strongly rooted in their place of residence. The social factor of local ties thus serves as the most important reason for the respondents to stay in the same place and to rebuild/refurbish there. This reason is followed, with 50.5%, by the fact that the respondents consider such an extreme flood very unlikely and therefore a move

unnecessary, which depends on the personal assessment of the potential flood risk. The belief that their building can be protected from flooding and therefore relocation is not necessary, which is also a personal perception of the situation, was checked much less frequently as a main reason, at 14.7%. Nearly one-fifth indicated that they did not have the energy or strength to move, whereupon it should be noted that this personal factor may also have a psychological component. The limited financial resources played a decisive role for 17.1% of the respondents, whereby both the personal financial situation, in general, could be poor as well as a sufficient use of the reconstruction funds could not be given. Merely 4.6% indicated they were unable to address the issue of settlement withdrawal/relocation during reconstruction, indicating that the vast majority were at least able to think about this issue. This personal reason may also have a psychological component, for example, if the individuals have not been able to deal with this fundamental issue due to psychological trauma. For about one seventh of the respondents, external factors also played a decisive role. In fact, 14.7% reported that they were unable to find suitable replacement areas nearby to build new housing, which in principle could be financed by the reconstruction fund. And 13.5% stated that they could not sell their building because the market value has dropped too much due to the flood. The least frequent answer, at 2.4%, was that the residential building was not affected by the 2021 flood—this low number is obviously due to the sample.

**What are key factors that influenced your decision to stay in the same location?**

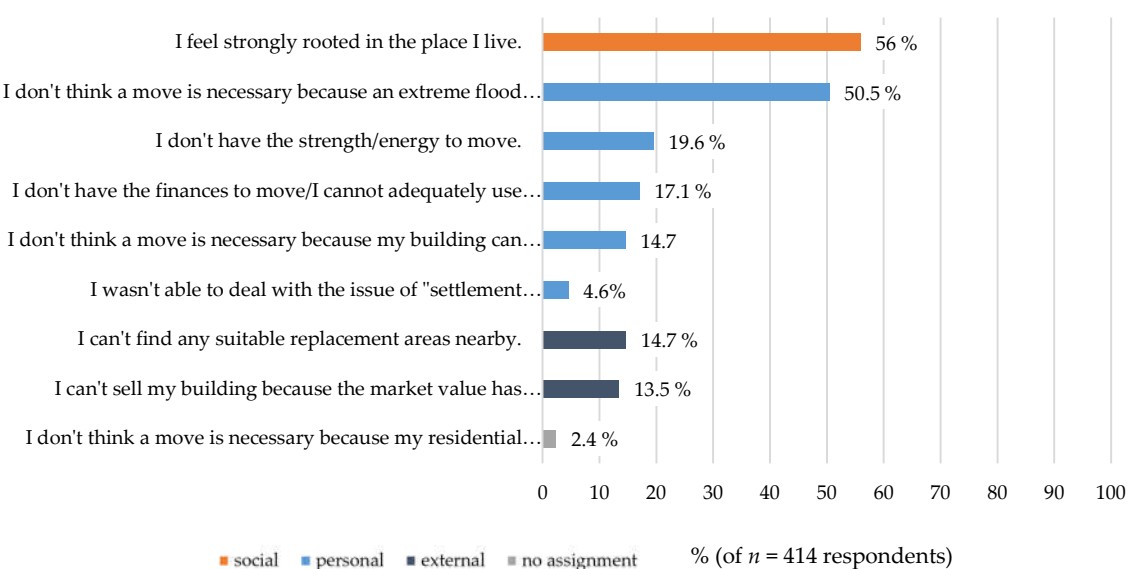

**Figure 7.** Decisive factors for staying in the same house/apartment as in July 2021, *n* = 414.

With regard to the decisive factors for staying on site, it is also worth examining whether there is a relationship between socioeconomic factors (age, gender, household income) and the reasons given (see Table A2). With respect to age, three correlations are significant. On the one hand, with increasing age, respondents more often have no energy/strength (anymore) for a move. In addition, they were more often unable to deal with the issue of relocation with increasing age. On the other hand, younger individuals more often indicated that they could not find suitable replacement areas nearby. In terms of the two different genders (female and male), 408 persons responded to both questions. The reason "I don't think a move is necessary because an extreme flood (like the July 2021 flood) is very rare.", was cited more often by men than women. A quite similar pattern emerges for the reason "I don't think a move is necessary because my building can be protected from floods". This is also consistent with previous findings regarding flood

risk/flood safety. For all other reasons, no statistically significant correlation is evident. Finally, any relationship with monthly net household income as an economic factor is again examined. Not even one statistically significant relationship was found between net household income classes and the indication of decisive reasons for staying. For instance, for the reason "I do not have the finances to move/ I cannot adequately use the reconstruction funds to move to another area." the significance is with $p = 0.086$ above the 5% threshold, although, for example, in the income class 900–1200€ the observed and expected frequencies differ by 7 to 2.8. However, it should be noted that all examined correlations with regard to the reasons for staying are weak ($0.1 \leq \varrho < 0.3$).

### 4.4. Settlement Retreat

In addition, all participants of the survey were also asked about their attitude towards some statements regarding the issue of settlement retreat, which is depicted in Figure 8. It is striking that the vast majority do not feel well informed regarding settlement retreat and the designation of the new floodplains. Thus, 65.7% do not agree (at all) with the statement that they feel well informed in this regard. More than half (51.6%), on the other hand, agree (completely) with the statement that they consider settlement retreat to be a useful tool in terms of risk prevention and climate adaptation. According to Spearman's correlation, neither tenants nor owners agree more strongly with this statement. However, there is a weak correlation with the perceived likelihood of one's house/apartment being affected by flooding again. The more likely the respondents consider their house/apartment to be affected by another flood, the more reasonable they personally also consider a settlement retreat in terms of risk prevention and climate adaptation. In addition, it is noticeable that 61.1% of those who had already moved or were planning to move agreed (completely) with the statement whereas only 47.8% of those who had decided to stay agreed (completely). Thus, often either the positive attitude towards settlement retreat is followed by action or the action leads to a more positive attitude toward this issue. Furthermore, 63.2% even agreed (completely) with the statement that sensitive or critical infrastructures such as hospitals or schools should be relocated from the immediate vicinity of the Ahr River, which has hardly played a role in the political discussion so far.

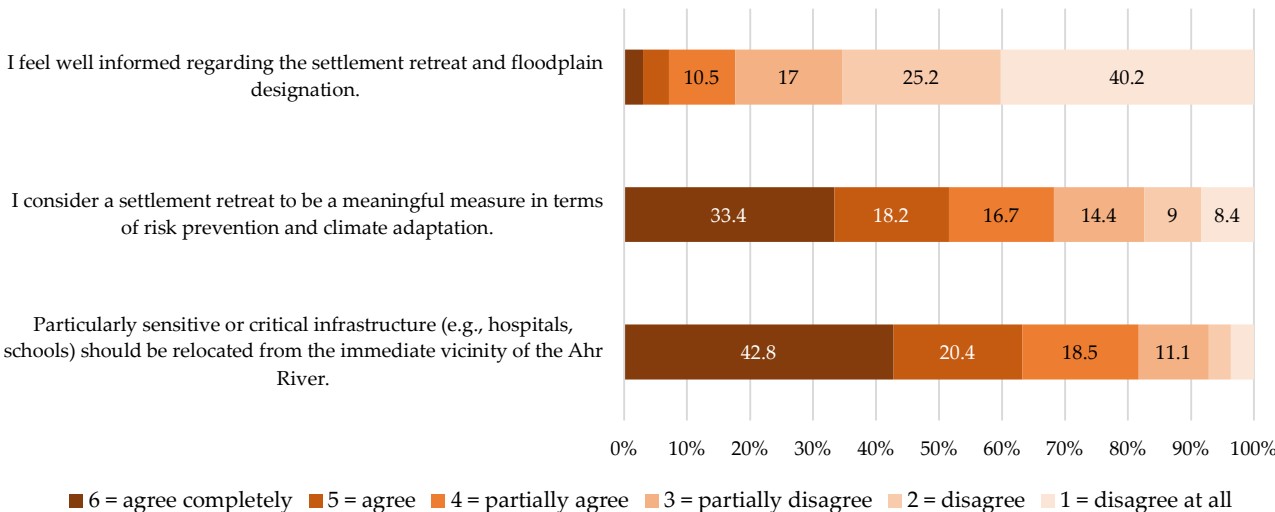

**Figure 8.** Agreement with statements regarding settlement retreat, relocation and floodplain designation on a six-point Likert scale (note: for clarity, all percentages less than 5% are not labelled).

## 5. Discussion

More than 40% of the respondents had to leave their accommodations after the July 2021 flood disaster. However, almost three-quarters of respondents (still or again) live in the same house/apartment as in July 2021 one year after the disaster and 12.9% still live in temporary accommodation/with friends/relatives. Furthermore, nearly one-fifth of the respondents already moved or at least plan to do so. Thus, a large proportion of respondents left their housing only temporarily. Ownership structures play a particularly important role regarding relocation, as tenants are significantly more likely to move. This is also consistent with other studies such as that conducted by the U.S. Census Bureau [45]. Among other things, this relationship between a tenancy and moving may be due to the fact that it is considerably less complicated for tenants to move, and they may feel less attached to the house or flat, as a tenancy is often not undertaken with the intention of living in the property forever. However, it may also be that landlords do not address reconstruction (sufficiently), and tenants, therefore, start searching for new residential properties. The fact that a tenancy is a decisive reason for moving is also confirmed by the fact that this reason was given by 43.5% of the people who already moved or want to move, making it the most frequently mentioned reason. In addition, a relationship could be found between staying/moving and place of residence, whereby respondents who lived within the legally designated floodplain of July 2021 are more inclined to move. However, this relationship should be treated with caution, as significantly more tenants also lived within the legally designated floodplain of July 2021. Beyond that, however, there is no relationship between staying/moving and the other sociodemographic factors of age, gender, and net monthly household income. The amount of damage is also not related to the decision to stay or leave, but the degree of recovery of the building/property at the time of the survey, which in turn is related to net household income. However, the fact that the amount of damage shows no relationship with the decision to stay/move may also be due to the fact that the damage is likely to be considerably higher for owners than for tenants, but the former are more likely to stay. It is also interesting to note that those reporting a low level of recovery for their building/property are more likely to live in temporary housing, and those living in temporary housing are also slightly more likely to consider a future moving. Net household income, on the other hand, does not affect whether people live in the same house or in temporary housing. That means, it cannot be stated that people with precarious financial situations are more likely to be forced to live in the same house, which may be badly damaged, nor that they are more likely to have to stay in temporary housing or with friends/relatives—the latter being probably mostly free of charge. For such a detailed investigation, further data would also be necessary, since it is possible that people with more money are more likely to be able to "make themselves at home" in the same house again, or that people with more money are more likely to be able to afford a decent vacation home. In addition, as other studies have already shown [35], the length of residence has an impact on moving or staying. Respondents who have lived in their apartment/house for a long time are slightly more likely to stay in it after the disaster.

In addition, respondents were asked directly about the reasons for staying/moving as well as the decisive factors when choosing a new location. But it should be kept in mind that the answers given here were predefined and that a statistical correlation cannot be equated with causality. In terms of reasons for moving, as already mentioned, a tenancy was cited most frequently by respondents, followed by the fact that the place of residence no longer offered/offers a quality of life. Thus, one year after the event, there is still a lack of leisure activities, sports activities, village stores, bakery stores, etc., and in some villages, entire streets are still destroyed. Over a quarter also said that the site was/is too dangerous for them—in relative terms, those aged 80 and over were particularly more likely to state this as a reason. The very elderly are also highly vulnerable in the event of a flood, as mobility is often limited in this age group. For instance, in Germany, 6.5% of those under 79 were in need of care in 2019, compared to 48.5% of those over 80 [46].

Nevertheless, no correlation with age, gender or net household income could be found for any of the cited reasons—except for the tenancy (young people gave this reason more frequently). This also means, for instance, that reconstruction is considered too expensive and/or too complex regardless of income, although one might expect lower-income individuals to cite this reason more often. In addition, those willing to relocate were also asked about the reasons that are decisive when choosing a new location. Many of the predefined reasons were rated as important or very important by the respondents, only the joint move with neighbors clearly stands out, as it is important or very important to only 11.3% of respondents. Interestingly, as age increases, the reason "sustainability and energy efficiency of the new building" becomes more important. This is consistent with other surveys that questioned people about their actual sustainable behavior [47] On the other hand, with regard to attitudes, e.g., in elections, the party "Buendnis 90/Die Gruenen", which stands for sustainability, in particular, performs especially well among younger people up to the age of 34 [48]. Therefore, one could expect younger people to be more concerned about issues like sustainability and energy efficiency. Looking at the two genders (as no one indicated divers), it is also noticeable that women tend to be less willing to take risks (flood-proof construction of the new building is more important to them) and are closer to their locality (short distance to their former place of residence and remaining in the municipality are also more important to them). In terms of household income, the only factor that stands out is "financial support from the state," which is for obvious reasons more important for lower incomes. The resettlement offer also played an important role in the aforementioned resettlement in the Eferding Basin in Austria [35], where, by contrast, resettlement was actively addressed by the state. In this case, the financial possibilities in particular initiated the process of consideration. Only if the resettlement was financially viable, the households considered more factors.

In terms of the reasons given by the group wishing to remain in their original place of residence, two internal factors are prominent in particular. More than half stated as a decisive factor that they feel strongly rooted in their place of residence as well as that they consider a move unnecessary since such an extreme flood only occurs very rarely. The latter reason was given significantly more often by men than by women, which is also in line with the question about the importance of factors in the choice of a new location, i.e., that women are less willing to take risks with regard to floods or resp. are more aware of them. In Austria, too, the personal assessment of flood risk played a role in the decision for or against resettlement [35]. Nevertheless, for about one seventh of the respondents in each case, the external factors also played a decisive role e.g., no suitable replacement areas nearby and a sharp decline in market prices. State intervention would be necessary to counter these external factors, especially in the Ahr Valley, which is very narrow and the areas that can be built are very limited. Possibly, an inter-communal land exchange system can provide a solution, as well as buying up buildings and land close to the river, so that those affected receive a fair price and the state in turn can create retention areas and space for the river. In the case of the very elderly respondents, the fact that they do not have the strength to move is also prominent—external help would likewise be conceivable for this internal factor. That elderly people have less strength or energy to move after the flood, is in line with general findings on moving behavior at different ages [49,50]. This is also supported by the fact that younger people more often reported that they could not find suitable replacement areas, i.e., they would be more willing to move— if the conditions were right. With regard to the reasons for staying, it is again noticeable that women are more aware of the risk of flooding. Overall, it is noticeable that among the reasons for staying, the respondents cited both "positive" reasons, such as strong rootedness, and "negative" reasons, such as lack of strength or lack of financial resources. Conversely, this also means that some people would not be averse to moving if they were not forced to stay in the area for "negative" reasons. To be specific, if, for example, they were not forced to stay on site due to limited financial and spatial resources. This could be counteracted by the state and local government decision-makers. On the one hand, by

using reconstruction funds and subsidy guidelines to encourage relocation in particular, and on the other hand, by designating areas in advance of flood disasters that are suitable as building sites and are located in a low-risk areas.

Regarding the settlement retreat and the designation of the new floodplains, a large percentage of respondents do not feel well informed, meaning that the communication between state agencies and the population is in need of improvement in this respect. Overall, the majority of those surveyed have a positive attitude towards settlement retreat, i.e., giving up and dismantling settlement structures, in the sense of risk prevention and climate change adaptation, although interestingly there is no difference between tenants and owners. Even more people are in favor of relocating particularly sensitive or critical infrastructures, such as hospitals or schools, from the immediate vicinity of the Ahr river. Unfortunately, many of these infrastructures are located near the river, e.g., nursing homes or rehabilitation clinics that are eager to advertise the great view and surroundings. Since a settlement retreat certainly meets with the general approval of the population, it should be discussed in detail with all those involved. This can also result in the population ultimately not agreeing when it becomes concrete and affects individuals directly. Nevertheless, this should be discussed publicly, and almost categorical exclusion of settlement retreat does not seem sensible from a scientific point of view.

## 6. Conclusions

Overall, a very differentiated picture emerges with regard to the topics of relocation and settlement retreat. The motivations for leaving or staying became apparent based on the household survey. In particular, place attachment and risk assessment are crucial for those staying, whereas a tenancy relationship is crucial for those moving. Furthermore, it became apparent that a considerable percentage of people had to leave their homes at least temporarily, and about one seventh had already moved away permanently. In some cases, people had only moved within their district, but about half of them had also moved completely to another city. It was also striking that a large number of respondents were not aware before the flood that they lived in a flood-prone area. The public authorities need to provide more information in order to create greater risk awareness. It would also be conceivable to make it compulsory to point out that a property or building is located in a flood-prone area, when purchasing it. Moreover, the survey clarified how those affected think about the issue of a settlement retreat. Overall, the respondents had a positive attitude towards this issue; in particular, the relocation of critical and sensitive infrastructures was perceived as a sensible means in terms of risk prevention and climate change adaptation. Thus, proactive relocation of the population from buildings at risk and of critical and sensitive infrastructure is a reasonable measure to reduce risk, and should be considered, especially in other high-risk regions that have not yet been hit by such a devastating disaster. However, the legal basis for this also needs to be adapted, as German regional planning and building law has so far been designed to control and strengthen settlement development. Thus, in the future, settlement retreat and dismantling of infrastructures need to be given a stronger position in the legal framework as well, so that state and local decision-makers are empowered to push relocation forward both before and after disasters. It would also be worth considering developing a general strategy for settlement retreat at the state level that could be used by individual municipalities, counties, or states.

In addition to the reasons given for or against a relocation, we also identified different groups that were, for example, more likely to relocate or more likely to still live in temporary housing one year after the event. However, in order to establish causalities, respectively to identify with confidence specific groups of people who are more likely to stay/relocate/live in temporary housing, further research is needed. For this purpose, in-depth personal interviews with those affected could help. Such in-depth interviews may also include, for example, network maps to better understand local ties and dynamics, as was done, for example, in the research in the Eferding Basin. Furthermore, one year after

the disaster, the relocation process is not yet complete, e.g., as some respondents are still living in temporary housing. Therefore, more movers could still be joining if the reconstruction takes too long or is not satisfactory, or if social structures decline as relocations increase (see Eferding Basin). For future research, it could also be interesting to find out how moving as well as staying on site affects the life satisfaction as well as the mental condition of the affected persons in the long run. For example, social ties can be disconnected by relocation, which in turn can have a negative impact on mental health. In addition, not all sociodemographic and physical variables were included in the present study. Thus, for future research, we recommend investigating further variables and their influence on relocation behavior, such as educational status, employment status, building types, or urban structure types. Thus, a follow-up survey is recommended, for example, with regard to local ties and mental health, as well as future research and investigation of further aspects such as educational status. In addition, since we have only dealt with relocation behavior at the household level, future research could address this issue at other levels as well. For example, one could examine the legal and policy perspectives on resettlement and relocation, as well as the possibilities for more effective resettlement due to climate change hazards.

**Supplementary Materials:** The following supporting information can be downloaded at: https://www.mdpi.com/article/10.3390/su15021407/s1, File S1: Household survey in the district of Ahrweiler on the flood event of July 2021—Excerpt of the questionnaire.

**Author Contributions:** Conceptualization, A.J.T. and J.B.; methodology, A.J.T. and A.J.; formal analysis, A.J.T.; investigation, A.J.T.; data curation, A.J.T.; writing—original draft preparation, A.J.T.; writing—review and editing, J.B., A.J., H.S. and A.J.T.; visualization, A.J.T. and H.S.; supervision, J.B.; project administration, J.B.; funding acquisition, J.B. All authors have read and agreed to the published version of the manuscript.

**Funding:** This research was funded by Federal Ministry of Education and Research/Bundesministerium für Bildung und Forschung (BMBF), grant number (FKZ) 01LR2102A.

**Institutional Review Board Statement:** The study was conducted in accordance with the Declaration of Helsinki, and approved by the Ethics Committee of the University of Stuttgart (protocol code (Az.) 22-017, 6 July 2022 (date of approval)).

**Informed Consent Statement:** Informed consent was obtained from all subjects involved in the study.

**Data Availability Statement:** Data is available in Appendix A.

**Acknowledgments:** We are very grateful for the funding provided by the Bundesministerium für Bildung und Forschung (BMBF) as part of the KAHR project. And we would like to sincerely thank the county of Ahrweiler for their great support in contacting the affected people and sending the invitation to them in order to participate in the study. We would also like to express our sincere appreciation to all participants who took the time to complete our questionnaire.

**Conflicts of Interest:** The authors declare no conflict of interest. The funders had no role in the design of the study; in the collection, analyses, or interpretation of data; in the writing of the manuscript; or in the decision to publish the results.

### Appendix A

Both the variables used and their frequencies or means (plus standard deviations) can be found in the Appendix A (Table A1), as well as the Spearman correlations addressed in the text (Table A2). In addition, two more figures are included to illustrate an issue mentioned above (Figure A1).

**Table A1.** Description of the variables used in the analysis.

| Variable | Definition | n | Summary Statistics Mean (Standard Dev.) or Percentages |
|---|---|---|---|
| Leaving home | Answer to the question of whether they had to leave their home because of a call, an official order, or because of the damage: yes = 1, no = 2. | 507 | 1 = 41.6%<br>2 = 57.4% |
| Duration of leaving home | Answer to the question "After how many days or months were you able to return to your home permanently?": after 1-2 days (1); after about a week (2); after about two weeks (3); after about one month (end of August 2021) (4); after about two months (end of September 2021) (5); after about four months (end of November 2021) (6); after about six months (end of January 2022) (7); after about eight months (end of March 2022) (8); not until today (9) | 216 | 1 = 6.0%<br>2 = 1.9%<br>3 = 2.3%<br>4 = 9.7%<br>5 = 11.1%<br>6 = 11.6%<br>7 = 7.9%<br>8 = 13.4%<br>9 = 36.1% |
| Current housing situation | Information on current housing situation (i.e., at the time of the survey) with several choices: in the same house/apartment as in July 2021 (1), in temporary housing/ with friends/ relatives (2), permanently in another house/apartment in the same district (3), permanently in another district (4), permanently in another city (5). | 512 | 1 = 73.0%<br>2 = 12.9%<br>3 = 4.1%<br>4 = 2.1%<br>5 = 7.8% |
| Current housing situation (excluding those who have already moved permanently) | Information on current housing situation (i.e., at the time of the survey) with several choices: in the same house/apartment as in July 2021 (1), in temporary housing/ with friends/ relatives (2) | 440 | 1 = 85.0%<br>2 = 15.0% |
| Location of the new residence | Location of the new building (if already moved permanently): inside legally defined floodplain valid in July 2021 (1); outside legally designated floodplain valid in July 2021, but within the July 2021 flood zone (2); outside legally designated floodplain and outside the July 2021 flood zone (3) | 53 | 1 = 11.3%<br>2 = 20.7%<br>3 = 67.9% |
| Future intended housing situation | Stay on site in the same house/apartment (1), move already carried out or planned (2) | 490 | 1 = 79.2%<br>2 = 20.8% |
| Move planned | Answer to the question if they plan to move: yes = 1, no = 2. | 422 | 1 = 7.1%<br>2 = 92.9% |
| Ownership structure (in July 2021) | Answer to the question whether they were living for rent or in their own property/ property of close relatives in July 2021: rent = 1, property/ property of close relatives = 2 | 515 | 1 = 32.4%<br>2 = 67.6% |
| Damage | Amount of financial damage to personal belongings and building/property: under 500€ (1), 500–999€ (2), 1000–4999€ (3), 5000–9999€ (4), 10,000–24,999€ (5), 25,000–49,999€ (6), 50,000–74,999€ (7), 75,000–99,999€ (8) and 100,000€ or more (9) | 492 | 1 = 0.2%<br>2 = 0.2%<br>3 = 5.7%<br>4 = 6.3%<br>5 = 12.0%<br>6 = 10.2%<br>7 = 8.9%<br>8 = 4.7%<br>9 = 52.8% |

| | | | |
|---|---|---|---|
| Current condition of the personal belongings and building/property | Comparison of current condition of personal belongings and building/property with pre-flood condition on a six-point Likert scale from 1 = completely replaced to 6 = significant deficiencies | 489 | 3.87 (1.67) |
| Duration of residence in house/apartment | Duration of residence in house/apartment of July 2021: in the house/apartment for 0 to 1.5 years (i.e., moved in in 2020 or 2021) (1), for 1.5 to 5.5 years (2), for 5.5 to 10.5 years (3), for 10.5 to 25.5 years (4), and more than 25.5 years (5) | 503 | 1 = 7.6% 2 = 18.3% 3 = 14.1% 4 = 32.2% 5 = 27.8% |
| Pre-flood risk awareness | Answer to the question of whether they knew they lived in a flood-prone area before the 2021 flood event: yes (1), no (2). | 496 | 1 = 17.7% 2 = 83.3% |
| Location of original residence | Location of the July 2021 residence: inside legally defined floodplain valid in July 2021 (1) or outside legally defined floodplain valid in July 2021 (2). *Note: A choice of five options was offered to the respondents in the questionnaire: inside the special hazard "yellow" zone; inside the legally defined floodplain of July 2021; outside the legally defined floodplain of July 2021 but inside the new floodplain provisionally defined by law; outside new floodplain provisionally defined by law but inside the flooded area from July 2021; outside new floodplain provisionally defined by law and outside flooded area from July 2021. However, since such a detailed classification was rather inconvenient for the evaluation, the categories were combined into two main categories.* | 401 | 1 = 25.7% 2 = 74.3% |
| Reasons for relocation | Answer to the question: "What are the reasons for your relocation?" (multiple answers were possible): I was living in a rented flat/house. (1); My place of residence no longer offers a quality of life. (2); The location is too dangerous for me. (3); Reconstruction/refurbishment is too expensive and/or too complex for me. (4) | 124 | 1 = 43.5% 2 = 40.3% 3 = 27.4% 4 = 10.5% |
| Importance of factors when deciding on a new location | Assessment of importance of different factors when deciding on a new location on a six-point Likert scale (from 1 = completely unimportant to 6 = very important): good social environment (1); flood-proof location of the new building (2); good connection to local supply (3); sustainability and energy efficiency of the new building (4); financial support from the state (5); good connection to public transport (6); flood-adapted construction of the new building (7); short distance to former place of residence (8); remaining in the same (associated) municipality (9); joint move with other affected neighbors (10). | 1–106 2–104 3–109 4–104 5–100 6–107 7–92 8–106 9–107 10–97 | 1 = 5.11 (1.12) 2 = 5.09 (1.41) 3 = 4.99 (1.29) 4 = 4.88 (1.37) 5 = 4.49 (1.74) 6 = 4.23 (1.69) 7 = 4.18 (1.56) 8 = 3.77 (1.82) 9 = 3.67 (2.07) 10 = 2.03 (1.50) |
| Importance of safety | Agreement with the thesis "I am someone to whom it is important to live in a safe environment and who avoids anything that threatens my own safety." on a six-point Likert scale from 1 = not true at all to 6 = absolutely true. | 504 | 4.30 (1.36) |
| Probability of a recurrence of flooding | Assessment of probability on a six-point Likert scale from 1 = very unlikely to 6 = very likely with regard to the following question: "How likely do you think it is that your current apartment/house will be affected by flooding again?" | 487 | 3.34 (1.56) |

| | | | |
|---|---|---|---|
| Severity of a recurrence of flooding | Assessment of severity on a six-point Likert scale from 1 = not bad to 6 = very bad with regard to the following question: "How do you assess the impact of a possible future flood on yourself personally?" | 493 | 4.91 (1.39) |
| Reasons for staying | Answer to the question: "What are key factors that influenced your decision to stay in the same location?" (multiple answers were possible): I feel strongly rooted in the place I live. (1); I don't think a move is necessary because an extreme flood is very rare. (2); I don't think a move is necessary because my building can be protected from floods. (3); I don't have the strength/energy to move. (4); I don't have the finances to move / I cannot adequately use the reconstruction funds to move to another area. (5); I wasn't able to deal with the issue of "settlement retreat/ relocation" during reconstruction. (6); I can't find any suitable replacement areas nearby. (7); I can't sell my building because the market value has dropped too much. (8); I don't think a move is necessary because my residential building was not affected by the 2021 flood. (9) | 414 | 1 = 56.0% <br> 2 = 50.5% <br> 3 = 14.7% <br> 4 = 19.6% <br> 5 = 17.1% <br> 6 = 4.6% <br> 7 = 14.7% <br> 8 = 13.5% <br> 9 = 2.4% |
| Feeling informed about the designation of the floodplains and the settlement retreat | Agreement with the statement "I feel well informed regarding the settlement retreat and floodplain designation." on a six-point Likert scale from 1 = disagree at all to 6 = agree completely. | 465 | 2.22 (1.34) |
| Assessment of a settlement retreat | Agreement with the statement "I consider a settlement retreat to be a meaningful measure in terms of risk prevention and climate change adaptation." on a six-point Likert scale from 1 = disagree at all to 6 = agree completely. | 479 | 4.28 (1.64) |
| Assessment of a relocation of critical and sensitive infrastructures | Agreement with the statement "Particularly sensitive or critical infrastructures (e.g., hospitals, schools) should be relocated from the immediate vicinity of the Ahr River." on a six-point Likert scale from 1 = disagree at all to 6 = agree completely. | 486 | 4.78 (1.38) |
| Gender | Female = 1, male = 2, divers = 3 | 510 | 1 = 47.6% <br> 2 = 52.4% <br> 3 = 0.0% |
| Age | Age groups from 20-29 years (1), from 30-49 years (2), from 50-64 years (3), from 65-79 years (4), and over 80 years (5) | 512 | 1 = 4.3% <br> 2 = 20.9% <br> 3 = 41.8% <br> 4 = 25.0% <br> 5 = 8.0% |
| Household income | Monthly net household income (after deduction of taxes and social contributions, but including all income, i.e., also income from rent, pension, etc.): below 900€ (1), 900–1299€ (2), 1300–1499€ (3), 1500–1999€ (4), 2000–2599€ (5), 2600–3599€ (6), 3600–4999€ (7) and 5000€ or more (8). | 443 | 1 = 1.1% <br> 2 = 5.4% <br> 3 = 4.1% <br> 4 = 11.5% <br> 5 = 18.5% <br> 6 = 24.2% <br> 7 = 16.9% <br> 8 = 18.3% |

**Table A2.** Spearman correlations studied in the analysis.

| Variables | n | Spearman's ϱ | *p* |
|---|---|---|---|
| Current housing situation (excluding those who have already moved permanently) & Move planned | 490 | −0.116 | 0.018 |
| Ownership structure (in July 2021) & Future intended housing situation | 489 | −0.323 | <0.001 |
| Gender & Future intended housing situation | 487 | 0.034 | 0.452 |
| Age & Future intended housing situation | 488 | −0.071 | 0.117 |
| Household income & Future intended housing situation | 423 | −0.067 | 0.167 |
| Current housing situation (excluding those who have already moved permanently) & Household income | 376 | 0.051 | 0.323 |
| Damage & Future intended housing situation | 468 | −0.063 | 0.173 |
| Current condition of the personal belongings and building/property & Household income | 443 | −0.154 | 0.002 |
| Current condition of the personal belongings and building/property & Future intended housing situation | 465 | 0.118 | 0.011 |
| Current condition of the personal belongings and building/property & Current housing situation (excluding those who have already moved permanently) | 418 | 0.293 | <0.001 |
| Ownership structure (in July 2021) & Current housing situation (excluding those who have already moved permanently) | 439 | 0.092 | 0.054 |
| Duration of residence in house/apartment & Future intended housing situation | 479 | −0.139 | 0.002 |
| Location of original residence & Pre-flood risk awareness | 391 | 0.214 | <0.001 |
| Pre-flood risk awareness & Future intended housing situation | 471 | −0.040 | 0.391 |
| Location of original residence & Future intended housing situation | 383 | −0.143 | 0.005 |
| Location of original residence & Ownership structure (in July 2021) | 400 | 0.313 | <0.001 |
| Age & I was living in a rented flat/house (reasons for relocation). | 122 | −0.215 | 0.018 |
| Age & The location is too dangerous for me (reasons for relocation). | 122 | 0.141 | 0.121 |
| Age & My place of residence no longer offers a quality of life (reasons for relocation). | 122 | 0.035 | 0.701 |
| Gender & I was living in a rented flat/house (reasons for relocation). | 121 | −0.034 | 0.714 |
| Gender & The location is too dangerous for me (reasons for relocation). | 121 | −0.139 | 0.129 |
| Gender & My place of residence no longer offers a quality of life (reasons for relocation). | 121 | 0.177 | 0.053 |
| Gender & Reconstruction/ refurbishment is too expensive/ too complex for me (reasons for relocation). | 121 | −0.053 | 0.566 |
| Household income & I was living in a rented flat/house (reasons for relocation). | 105 | −0.140 | 0.154 |
| Household income & The location is too dangerous for me (reasons for relocation). | 105 | 0.169 | 0.085 |
| Household income & My place of residence no longer offers a quality of life (reasons for relocation). | 105 | 0.046 | 0.644 |
| Household income & Reconstruction/ refurbishment is too expensive/ too complex for me (reasons for relocation). | 105 | −0.049 | 0.622 |
| Gender & Importance of safety | 500 | −0.047 | 0.292 |
| Gender & Probability of a recurrence of flooding | 482 | −0.136 | 0.003 |
| Gender & Severity of a recurrence of a flooding | 488 | −0.173 | <0.001 |
| Age & Reasons for staying: Rootedness (1). | 410 | −0.112 | 0.023 |
| Age & Reasons for staying: Rarity of an extreme flood (2). | 410 | 0.010 | 0.839 |
| Age & Reasons for staying: Building can be protected (3). | 410 | 0.021 | 0.670 |
| Age & Reasons for staying: No strength/energy (4). | 410 | 0.157 | 0.001 |
| Age & Reasons for staying: No finances (5). | 410 | 0.025 | 0.620 |
| Age & Reasons for staying: Not able to deal with the issue (6). | 410 | 0.103 | 0.038 |

| | | | |
|---|---|---|---|
| Age & Reasons for staying: No suitable replacement areas (7). | 410 | −0.136 | 0.006 |
| Age & Reasons for staying: Not able to sell due to too much decreased market price (8). | 410 | 0.030 | 0.539 |
| Age & Reasons for staying: Building was not affected (9). | 410 | 0.003 | 0.946 |
| Gender & Reasons for staying: Rootedness (1). | 408 | −0.073 | 0.139 |
| Gender & Reasons for staying: Rarity of an extreme flood (2). | 408 | 0.210 | <0.001 |
| Gender & Reasons for staying: Building can be protected (3). | 408 | 0.118 | 0.017 |
| Gender & Reasons for staying: No strength/ energy (4). | 408 | −0.015 | 0.770 |
| Gender & Reasons for staying: No finances (5). | 408 | 0.086 | 0.082 |
| Gender & Reasons for staying: Not able to deal with the issue (6). | 408 | 0.041 | 0.404 |
| Gender & Reasons for staying: No suitable replacement areas (7). | 408 | −0.019 | 0.701 |
| Gender & Reasons for staying: Not able to sell due to too much decreased market price (8). | 408 | −0.055 | 0.269 |
| Gender & Reasons for staying: Building was not affected (9). | 408 | −0.036 | 0.464 |
| Household income &Reasons for staying: Rootedness (1). | 355 | −0.003 | 0.957 |
| Household income & Reasons for staying: Rarity of an extreme flood (2). | 355 | 0.090 | 0.091 |
| Household income & Reasons for staying: Building can be protected (3). | 355 | 0.093 | 0.081 |
| Household income & Reasons for staying: No strength/ energy (4). | 355 | −0.093 | 0.081 |
| Household income & Reasons for staying: No finances (5). | 355 | −0.091 | 0.086 |
| Household income & Reasons for staying: Not able to deal with the issue (6). | 355 | 0.021 | 0.692 |
| Household income & Reasons for staying: No suitable replacement areas (7). | 355 | 0.080 | 0.132 |
| Household income & Reasons for staying: Not able to sell due to too much decreased market price (8). | 355 | 0.038 | 0.472 |
| Household income & Reasons for staying: Building was not affected (9). | 355 | 0.035 | 0.510 |
| Ownership structure (in July 2021) & Assessment of a settlement retreat | 478 | −0.075 | 0.100 |
| Probability of a recurrence of flooding & Assessment of a settlement retreat | 456 | 0.176 | <0.001 |

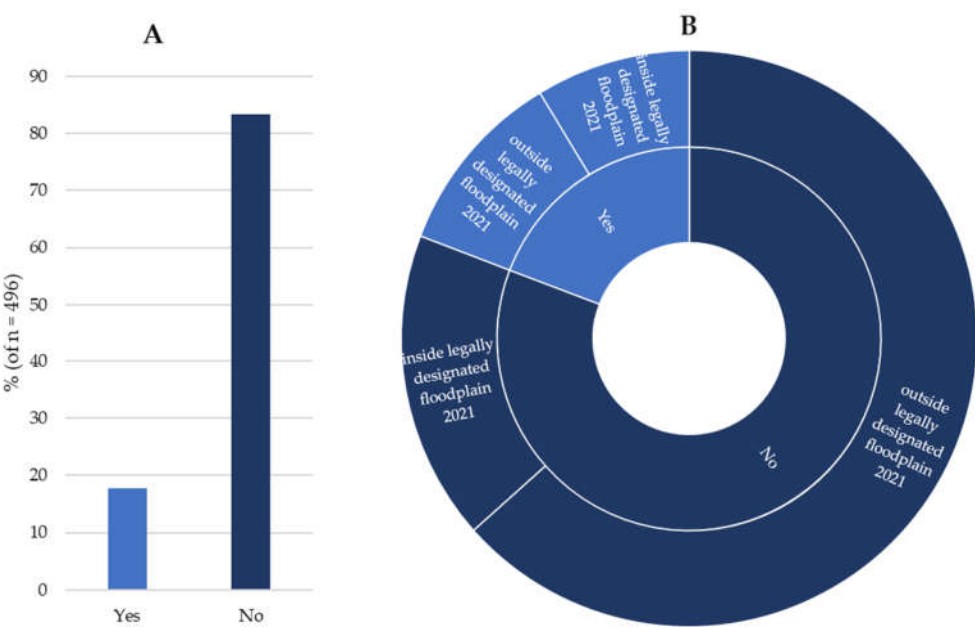

**Figure A1.** (**A**) Percentage of people who knew or did not know they lived in a flood-prone area before the 2021 flood event; (**B**) Distribution of the location of the residential building (outside or inside the legally designated floodplain valid in 2021) on the knowledge/non-knowledge of living in a flood-prone area before the 2021 flood event.

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
