# Peer review of "Adaptation after Extreme Flooding Events: Moving or Staying? The Case of the Ahr Valley in Germany"

_sustainability, doi:10.3390/su15021407_

Round 1
Reviewer 1 Report
The paper is interesting and addresses an important issue. This paper has a well-organized structure. A comprehensive presentation of the results and discussion has been provided.
Several minor concerns need to be addressed:
This abstract is unnecessarily long and contains many basic details. It should be shorter.
The methodology of the research has not been explained clearly. Please revise this section.
Please discuss how this study can facilitate the developing an evacuation planning system in the literature. Please see:
"An integrated decision model for managing hospital evacuation in response to an extreme flood event: a case study of the Hawkesbury‐Nepean River, NSW, Australia." Safety science 155 (2022): 105867.
"A modelling framework to design an evacuation support system for healthcare infrastructures in response to major flood events." Progress in disaster science 13 (2022): 100218.
Reviewer 2 Report
Dear Authors,
In line with the proofreading criteria of the publisher, I prepared a report, which would be as follows:
The content of the proposed paper meets the objectives set out in the special issue information letter.
Using the scientific methods applied in accordance with the author’s research objectives resulted useful scientific achievements.
The strength of the article is, among other things that the authors analysed and evaluated - through a specific case study - the behaviour of the population affected by the major effects of flash flood disaster and the background of their decisions regarding leaving the endangered areas.
In addition to the excellent methodological research work and literature assessment complemented by the authors, I recommend examining in 1-st or 2-nd chapters the national-level initiatives for solving a similar situations, the points of connection with international and European initiatives, such as the Sendai framework for disaster risk reduction, or the European civil protection mechanism.
The authors state at the end of Chapter 1 (lines 58-60) that the findings can also help to guide and modify reconstruction policies, such as relocation and development of alternative settlement sites. In my opinion it is also would be recommended that, in Chapter 5, the authors provide some general proposals for the possible modification of local, regional or national restoration strategies or related legal regulations.
In addition, it should also be also considered that the authors give their suggestions to state and local government decision-makers in order to prevent similar events and prepare for the elimination of such a catastrophic consequences. For example, the preliminary assessment of potential endangered areas, the development of monitoring and public alarm systems, and the preliminary resettlement of the population from endangered buildings can be the appropriate solutions.
It can also be important to determine the directions of further research.
The references used in the main chapters are relevant and assist the reader to understand the authors proposals. A small suggestion is that the authors write out the abbreviation “SPSS” the first time when it is used.
The illustrations used are regular and they significantly assist the reader in understanding the authors' investigation results and findings.
Based on the above, I suggest publishing the reviewed article.
Reviewer 3 Report
Dear Authors,
The manuscript 'Adaptation after extreme events: Moving or staying?' is adequate to be published in the Special Issue Climate Change, Adaptation and Disaster Risk Reduction – Planning Perspectives of the Journal Sustainability.
My comments are formulated constructively to improve the quality of this interesting and timely manuscript.
Line 2: The title needs to be rewritten to reflect the manuscript's content better. I suggest 'Adaptation after extreme flooding events: Moving or staying? The case of Ahr Valley in Germany'.
Lines 7 – 8: "Extreme precipitation was influenced by climate change". Do you have any citations/references for this attribution? Please cite studies on the rainfall projections for Germany for context. In addition, add and cite studies on intense rainfall trends in Germany. Also, it would be essential to include a sentence about the projection of intense rainfall in Germany in the introduction section.
Unfortunately, this research did not look at the relationship between the education of the respondents (secondary school, university, etc.) and the choice of reasons. This should be written in the manuscript as one of the research/methodological limitations. It would be imperative to assess the importance of the diverse levels of education in the decision of moving or staying and the statistical significance.
Lines 348 and 551: Each number from 1 to 6 should have a description category clearly written in the manuscript. Such as ‘no answer’ and from ‘strongly disagree’ to ‘strongly agree’.
Lines 507 – 508: The green bars are wrongly represented as 'psychological'. Please use a different classification instead of 'psychological'.
The survey questions must be included in the supplementary information.
Best wishes,
Reviewer
